# TOWARD EVALUATING ROBUSTNESS OF DEEP REINFORCEMENT LEARNING WITH CONTINUOUS CONTROL

**Tsui-Wei Weng**[1,*]**, Krishnamurthy (Dj) Dvijotham**[2,♣]**, Jonathan Uesato**[2,♣]**, Kai Xiao**[1,♣,*]
**Sven Gowal**[2,♣]**, Robert Stanforth**[2,♣]**, Pushmeet Kohli**[2]
[1]MIT, [2]DeepMind
twweng@mit.edu, {dvij,juesato}@google.com, kaix@mit.edu,
{sgowal,stanforth,pushmeet}@google.com

## ABSTRACT

Deep reinforcement learning has achieved great success in many previously difficult reinforcement learning tasks, yet recent studies show that deep RL agents are also unavoidably susceptible to adversarial perturbations, similar to deep neural networks in classification tasks. Prior works mostly focus on model-free adversarial attacks and agents with discrete actions. In this work, we study the problem of continuous control agents in deep RL with adversarial attacks and propose the first two-step algorithm based on learned model dynamics. Extensive experiments on various MuJoCo domains (Cartpole, Fish, Walker, Humanoid) demonstrate that our proposed framework is much more effective and efficient than model-free attacks baselines in degrading agent performance as well as driving agents to unsafe states.

## 1 INTRODUCTION

Deep reinforcement learning (RL) has revolutionized the fields of AI and machine learning over the last decade. The introduction of deep learning has achieved unprecedented success in solving many problems that were intractable in the field of RL, such as playing Atari games from pixels and performing robotic control tasks (Mnih et al., 2015; Lillicrap et al., 2015; Tassa et al., 2018). Unfortunately, similar to the case of deep neural network classifiers with adversarial examples, recent studies show that deep RL agents are also vulnerable to adversarial attacks.

A commonly-used threat model allows the adversary to manipulate the agent's observations at every time step, where the goal of the adversary is to decrease the agent's total accumulated reward. As a pioneering work in this field, (Huang et al., 2017) show that by leveraging the FGSM attack on each time frame, an agent's average reward can be significantly decreased with small input adversarial perturbations in five Atari games. (Lin et al., 2017) further improve the efficiency of the attack in (Huang et al., 2017) by leveraging heuristics of detecting a good time to attack and luring agents to bad states with sample-based Monte-Carlo planning on a trained generative video prediction model.

Since the agents have discrete actions in Atari games (Huang et al., 2017; Lin et al., 2017), the problem of attacking Atari agents often reduces to the problem of finding adversarial examples on image classifiers, also pointed out in (Huang et al., 2017), where the adversaries intend to craft the input perturbations that would drive agent's new action to deviate from its nominal action. However, for agents with continuous actions, the above strategies can not be directly applied. Recently, (Uesato et al., 2018) studied the problem of adversarial testing for continuous control domains in a similar but slightly different setting. Their goal was to efficiently and effectively find catastrophic failure given a trained agent and to predict its failure probability. The key to success in (Uesato et al., 2018) is the availability of agent training history. However, such information may not always be accessible to the users, analysts, and adversaries.

Besides, although it may not be surprising that adversarial attacks exist for the deep RL agents as adversarial attacks have been shown to be possible for neural network models in various supervised

---

*Work done during summer internship at DeepMind, UK. ♣ Equal contributions.

learning tasks. However, the vulnerability of RL agents can not be easily discovered by existing baselines which are model-free and build upon random searches and heuristics – this is also verified by our extensive experiments on various domains (e.g. walker, humanoid, cartpole, and fish), where the agents still achieve close to their original best rewards even with baseline attacks at every time step. Hence it is important and necessary to have a systematic methodology to design *non-trivial* adversarial attacks, which can *efficiently* and *effectively* discover the vulnerabilities of deep RL agents – this is indeed the motivation of this work.

This paper takes a first step toward this direction by proposing the first sample-efficient model-based adversarial attack. Specifically, we study the robustness of deep RL agents in a more challenging setting where the agent has continuous actions and its training history is not available. We consider the threat models where the adversary is allowed to manipulate an agent's observations or actions with small perturbations, and we propose a two-step algorithmic framework to find efficient adversarial attacks based on learned dynamics models. Experimental results show that our proposed model-based attack can successfully degrade agent performance and is also more effective and efficient than model-free attacks baselines.

The contributions of this paper are the following:

- To the best of our knowledge, we propose the first model-based attack on deep RL agents with continuous actions. Our proposed attack algorithm is a general two-step algorithm and can be directly applied to the two commonly-used threat models (observation manipulation and action manipulation).

- We study the efficiency and effectiveness of our proposed *model-based* attack with model-free attack baselines based on random searches and heuristics. We show that our model-based attack can degrade agent performance in numerous MuJoCo domains by up to $4\times$ in terms of total reward and up to $4.6\times$ in terms of distance to unsafe states (smaller means stronger attacks) compared to the model-free baselines.

- Our proposed *model-based* attack also outperform all the baselines by a large margin in a *weaker* adversary setting where the adversary cannot attack at every time step. In addition, ablation study on the effect of planning length in our proposed technique suggests that our method can still be effective even when the learned dynamics model is not very accurate.

## 2 BACKGROUND

**Adversarial attacks in reinforcement learning.** Compared to the rich literature of adversarial examples in image classifications (Szegedy et al., 2013) and other applications (including natural language processing (Jia & Liang, 2017), speech (Carlini & Wagner, 2018), etc), there is relatively little prior work studying adversarial examples in deep RL. One of the first several works in this field are (Huang et al., 2017) and (Lin et al., 2017), where both works focus on deep RL agent in Atari games with pixels-based inputs and discrete actions. In addition, both works assume the agent to be attacked has accurate policy and the problem of finding adversarial perturbation of visual input reduces to the same problem of finding adversarial examples on image classifiers. Hence, (Huang et al., 2017) applied FGSM (Goodfellow et al., 2015) to find adversarial perturbations and (Lin et al., 2017) further improved the efficiency of the attack by heuristics of observing a good timing to attack – when there is a large gap in agents action preference between most-likely and least-likely action. In a similar direction, (Uesato et al., 2018) study the problem of adversarial testing by leveraging rejection sampling and the agent training histories. With the availability of training histories, (Uesato et al., 2018) successfully uncover bad initial states with much fewer samples compared to conventional Monte-Carlo sampling techniques. Recent work by (Gleave et al., 2019) consider an alternative setting where the agent is attacked by another agent (known as adversarial policy), which is different from the two threat models considered in this paper. Finally, besides adversarial attacks in deep RL, a recent work (Wang et al., 2019) study verification of deep RL agent under attacks, which is beyond the scope of this paper.

**Learning dynamics models.** Model-based RL methods first acquire a predictive model of the environment dynamics, and then use that model to make decisions (Atkeson & Santamaria, 1997). These model-based methods tend to be more sample efficient than their model-free counterparts,

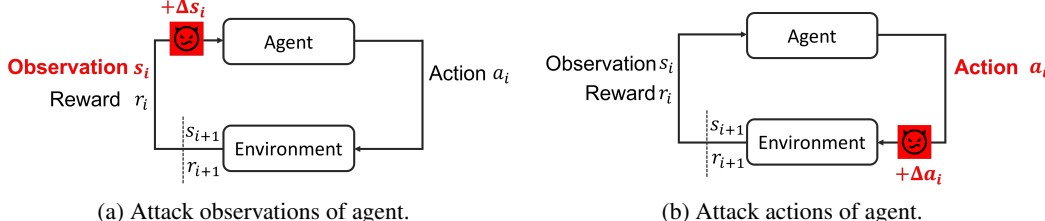

(a) Attack observations of agent.   (b) Attack actions of agent.

Figure 1: Two commonly-used threat models.

and the learned dynamics models can be useful across different tasks. Various works have focused on the most effective ways to learn and utilize dynamics models for planning in RL (Kurutach et al., 2018; Chua et al., 2018; Chiappa et al., 2017; Fu et al., 2016).

## 3 PROPOSED FRAMEWORK

In this section, we first describe the problem setup and the two threat models considered in this paper. Next, we present an algorithmic framework to rigorously design adversarial attacks on deep RL agents with continuous actions.

### 3.1 PROBLEM SETUP AND FORMULATION

Let $s_i \in \mathbb{R}^N$ and $a_i \in \mathbb{R}^M$ be the observation vector and action vector at time step $i$, and let $\pi : \mathbb{R}^N \to \mathbb{R}^M$ be the deterministic policy (agent). Let $f : \mathbb{R}^N \times \mathbb{R}^M \to \mathbb{R}^N$ be the dynamics model of the system (environment) which takes current state-action pair $(s_i, a_i)$ as inputs and outputs the next state $s_{i+1}$. We are now in the role of an adversary, and as an adversary, our goal is to drive the agent to the (un-safe) target states $s_{\text{target}}$ within the $\epsilon$ budget constraints.

We can formulate this goal into two optimization problems, as we will illustrate shortly below. Within this formalism, we can consider two threat models:

**Threat model (i): Observation manipulation.** For the threat model of observation manipulation, an adversary is allowed to manipulate the observation $s_i$ that the agent perceived within an $\epsilon$ budget:

$$\|\Delta s_i\|_\infty \leq \epsilon, \quad L_s \leq s_i + \Delta s_i \leq U_s, \tag{1}$$

where $\Delta s_i \in \mathbb{R}^N$ is the crafted perturbation and $U_s \in \mathbb{R}^N, L_s \in \mathbb{R}^N$ are the observation limits.

**Threat model (ii): Action manipulation.** For the threat model of action manipulation, an adversary can craft $\Delta a_i \in \mathbb{R}^M$ such that

$$\|\Delta a_i\|_\infty \leq \epsilon, \quad L_a \leq a_i + \Delta a_i \leq U_a, \tag{2}$$

where $U_a \in \mathbb{R}^M, L_a \in \mathbb{R}^M$ are the limits of agent's actions.

**Our formulations.** Given an initial state $s_0$ and a pre-trained policy $\pi$, our (adversary) objective is to minimize the total distance of each state $s_i$ to the pre-defined target state $s_{\text{target}}$ up to the unrolled (planning) steps $T$. This can be written as the following optimization problems in Equations 3 and 4 for the **Threat model (i)** and **(ii)** respectively:

$$\min_{\Delta s_i} \quad \sum_{i=1}^{T} d(s_i, s_{\text{target}}) \tag{3}$$

$$\text{s.t.} \quad a_i = \pi(s_i + \Delta s_i), \ s_{i+1} = f(s_i, a_i), \ \text{Constraint (1)}, \ i \in \mathbb{Z}_T,$$

$$\min_{\Delta a_i} \quad \sum_{i=1}^{T} d(s_i, s_{\text{target}}) \tag{4}$$

$$\text{s.t.} \quad a_i = \pi(s_i), \ s_{i+1} = f(s_i, a_i + \Delta a_i), \ \text{Constraint (2)}, \ i \in \mathbb{Z}_T.$$

A common choice of $d(x, y)$ is the squared $\ell_2$ distance $\|x - y\|_2^2$ and $f$ is the learned dynamics model of the system, and $T$ is the unrolled (planning) length using the dynamics models.

## 3.2 OUR ALGORITHM

In this section, we propose a two-step algorithm to solve Equations 3 and 4. The core of our proposal consists of two important steps: learn a dynamics model $f$ of the environment and deploy optimization technique to solve Equations 3 and 4. We first discuss the details of each factor, and then present the full algorithm by the end of this section.

**Step 1: learn a good dynamics model $f$.** Ideally, if $f$ is the exact (perfect) dynamics model of the environment and assuming we have an optimization oracle to solve Equations 3 and 4, then the solutions are indeed the optimal adversarial perturbations that give the minimal total loss with $\epsilon$-budget constraints. Thus, learning a good dynamics model can conceptually help on developing a strong attack. Depending on the environment, different forms of $f$ can be applied. For example, if the environment of concerned is close to a linear system, then we could let $f(s, a) = As + Ba$, where $A$ and $B$ are unknown matrices to be learned from the sample trajectories $(s_i, a_i, s_{i+1})$ pairs. For a more complex environment, we could decide if we still want to use a simple linear model (the next state prediction may be far deviate from the true next state and thus the learned dynamical model is less useful) or instead switch to a non-linear model, e.g. neural networks, which usually has better prediction power but may require more training samples. For either case, the model parameters $A, B$ or neural network parameters can be learned via standard supervised learning with the sample trajectories pairs $(s_i, a_i, s_{i+1})$.

**Step 2: solve Equations 3 and 4.** Once we learned a dynamical model $f$, the next immediate task is to solve Equation 3 and 4 to compute the adversarial perturbations of observations/actions. When the planning (unrolled) length $T > 1$, Equation 3 usually can not be directly solved by off-the-shelf convex optimization toolbox since the deel RL policy $\pi$ is usually a non-linear and non-convex neural network. Fortunately, we can incorporate the two equality constraints of Equation 3 into the objective and with the remaining $\epsilon$-budget constraint (Equation 1), Equation 3 can be solved via projected gradient descent (PGD) [1]. Similarly, Equation 4 can be solved via PGD to get $\Delta a_i$. We note that, similar to the $n$-step model predictive control, our algorithm could use a much larger planning (unrolled) length $T$ when solving Equations 3 and 4 and then only apply the first $n \ (\leq T)$ adversarial perturbations on the agent over $n$ time steps. Besides, with the PGD framework, $f$ is not limited to feed-forward neural networks. Our proposed attack is summarized in Algorithm 2 for **Step 1**, and Algorithm 3 for **Step 2**.

---

**Algorithm 1** Collect_trajectories

---

1: **Input:** pre-trained policy $\pi$, MaxSampleSize $n_s$, environment `env`
2: **Output:** a set of trajectory pairs $\mathcal{S}$
3: $k \leftarrow 0, \mathcal{S} \leftarrow \phi$
4: $s_0 \leftarrow$ `env`.reset()
5: **while** $k < n_s$ **do**
6: $\quad a_k \leftarrow \pi(s_k)$
7: $\quad s_{k+1} \leftarrow$ `env`.step($a_k$)
8: $\quad \mathcal{S} \cup \{(s_k, a_k, s_{k+1})\}$
9: $\quad k \leftarrow k + 1$
10: **end while**
11: **Return** $\mathcal{S}$

---

## 4 EXPERIMENTS

In this section, we conduct experiments on standard reinforcement learning environment for continuous control (Tassa et al., 2018). We demonstrate results on 4 different environments in Mu-

---

[1]Alternatively, standard optimal control methods such as Linear Quadratic Regulator (LQR) and iterative Linear Quadratic Regulator (i-LQR) can also be applied to solve Equations 3 and 4 approximately.

---

**Algorithm 2** learn_dynamics

---

1: **Input:** pre-trained policy $\pi$, MaxSampleSize $n_s$, environment `env`, trainable parameters $W$
2: **Output:** learned dynamical model $f(s, a; W)$
3: $\mathcal{S}_{\text{agent}} \leftarrow$ **Collect_trajectories**$(\pi, n_s, \text{env})$
4: $\mathcal{S}_{\text{random}} \leftarrow$ **Collect_trajectories**$(\text{random\_policy}, n_s, \text{env})$
5: $f(s, a; W) \leftarrow$ supervised_learning_algorithm$(\mathcal{S}_{\text{agent}} \cup \mathcal{S}_{\text{random}}, W)$
6: **Return** $f(s, a; W)$

---

**Algorithm 3** model_based_attack

---

1: **Input:** pre-trained policy $\pi$, learned dynamical model $f(s, a; W)$, threat model, maximum perturbation magnitude $\epsilon$, unroll length $T$, apply perturbation length $n$ ($\leq T$)
2: **Output:** a sequence of perturbation $\delta_1, \ldots, \delta_n$
3: **if** threat model is observation manipulation (Eq. 1) **then**
4:    Solve Eq. 3 with parameters $(\pi, f, \epsilon, T)$ via PGD to get $\delta_1, \ldots, \delta_T$
5: **else if** threat model is action manipulation (Eq. 2) **then**
6:    Solve Eq. 4 with parameters $(\pi, f, \epsilon, T)$ via PGD to get $\delta_1, \ldots, \delta_T$
7: **end if**
8: **Return** $\delta_1, \ldots, \delta_n$

---

JoCo Tassa et al. (2018) and corresponding tasks: Cartpole-balance/swingup, Fish-upright, Walker-stand/walk and Humanoid-stand/walk. For the deep RL agent, we train a state-of-the-art D4PG agent (Barth-Maron et al., 2018) with default Gaussian noise $\mathcal{N}(\mathbf{0}, \mathbf{0.3I})$ on the action and the score of the agents without attacks is summarized in Appendix A.3. The organization is as follows: we first evaluate the effectiveness of our proposed model-based attack and three model-free baselines in terms of both loss and reward. Next, we conduct ablation study on the key parameter of our algorithm the planning length $T$, evaluate our algorithm on a weaker attack setting and also discuss the efficiency of our proposed attack in terms of sample complexity.

**Evaluations.** We conduct experiments for 10 different runs, where the environment is reset to different initial states in different runs. For each run, we attack the agent for one episode with 1000 time steps (the default time intervals is usually 10 ms) and we compute the total loss and total return reward. The total loss calculates the total distance of current state to the unsafe states and the total return reward measures the true accumulative reward from the environment based on agent's action. Hence, the attack algorithm is stronger if the total return reward and the total loss are smaller.

**Baselines.** We compare our algorithm with the following model-free attack baselines with random searches and heuristics:

- **rand-U**: generate $m$ randomly perturbed trajectories from Uniform distribution with interval $[-\epsilon, \epsilon]$ and return the trajectory with the smallest loss (or reward),
- **rand-B**: generate $m$ randomly perturbed trajectories from Bernoulli distribution with probability $1/2$ and interval $[-\epsilon, \epsilon]$, and return the trajectory with the smallest loss (or reward),
- **flip**: generate perturbations by flipping agent's observations/actions within the $\epsilon$ budget in $\ell_\infty$ norm.

For **rand-U** and **rand-B**, they are similar to Monte-Carlo sampling methods, where we generate $m$ sample trajectories from random noises and report the loss/reward of the best trajectory (with minimum loss or reward among all the trajectories). We set $m = 1000$ throughout the experiments. More details see Appendix A.2.

**Our algorithm.** A 4-layer feed-forward neural network with 1000 hidden neurons per layer is trained as the dynamics model $f$ respectively for the domains of Cartpole, Fish, Walker and Humanoid. We use standard $\ell_2$ loss (without regularization) to learn a dynamics model $f$. Instead of using recurrent neural network to represent $f$, we found that the 1-step prediction for dynamics with the 4-layer feed-forward network is already good for the MuJoCo domains we are studying. Specifically, for the Cartpole and Fish, we found that 1000 episodes ($1e6$ training points) are sufficient

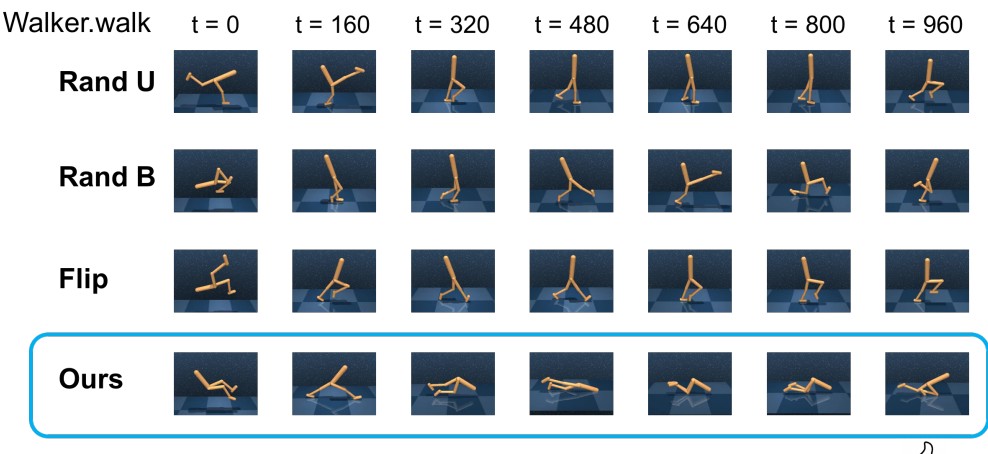

Figure 2: Video frames of best attacks in each baseline among 10 runs for the Walker.walk example. Only our proposed attack can constantly make the Walker fall down (since we are minimizing its head height to be zero).

to train a good dynamics model (the mean square error for both training and test losses are at the order of $10^{-5}$ for Cartpole and $10^{-2}$ for Fish), while for the more complicated domain like Walker and Humanoid, more training points ($5e6$) are required to achieve a low test MSE error (at the order of $10^{-1}$ and $10^0$ for Walker and Humanoid respectively). Consequently, we use larger planning (unrolled) length for Cartpole and Fish (e.g. $T = 10, 20$), while a smaller $T$ (e.g. 3 or 5) is used for Walker and Humanoid. Meanwhile, we focus on applying projected gradient descent (PGD) to solve Equation 3 and 4. We use Adam as the optimizer with optimization steps equal to 30 and we report the best result for each run from a combination of 6 learning rates, 2 unroll length $\{T_1, T_2\}$ and $n$ steps of applying PGD solution with $n \leq T_i$.

### 4.1 RESULTS

For observation manipulation, we report the results on Walker, Humanoid and Cartpole domains with tasks (stand, walk, balance, swingup) respectively. The unsafe states $s_{\text{target}}$ for Walker and Humanoid are set to be zero head height, targeting the situation of falling down. For Cartpole, the unsafe states are set to have $180°$ pole angle, corresponding to the cartpole not swinging up and nor balanced. For the Fish domain, the unsafe states for the upright task target the pose of swimming fish to be not upright, e.g. zero projection on the $z$-axis.

The full results of both two threat models on observation manipulation and action manipulation are shown in Table 1a, b and c, d respectively. Since the loss is defined as the distance to the target (unsafe) state, the lower the loss, the stronger the attack. It is clear that our proposed attack achieves much lower loss in Table 1a & c than the other three model-free baselines, and the averaged ratio is also listed in 1b & d. Notably, over the 10 runs, our proposed attack always outperforms baselines for the threat model of observation perturbation and the Cartpole domain for the threat model of action perturbation, while still superior to the baselines despite losing two times to the **flip** baseline on the Fish domain.

To have a better sense on the numbers, we give some quick examples below. For instance, as shown in Table 1a and b, we show that the average total loss of walker head height is almost unaffected for the three baselines – if the walker successfully stand or walk, its head height usually has to be greater than $1.2$ at every time step, which is $1440$ for one episode – while our attack can successfully lower the walker head height by achieving an average of total loss of $258(468)$, which is roughly $0.51(0.68)$ per time step for the stand (walk) task. Similarly, for the humanoid results, a successful humanoid usually has head height greater than $1.4$, equivalently a total loss of $1960$ for one episode, and Table 1a shows that the d4pg agent is robust to the perturbations generated from the three model-free baselines while being vulnerable to our proposed attack. Indeed, as shown in Figure 2, the

Table 1: Compare three model-free attack baselines (**rand-U**, **rand-B**, **flip**) and our algorithm (**Ours**) in 4 different domains and tasks. We report the following statistics over 10 different runs: mean, standard deviation, averaged ratio, and best attack (number of times having smallest loss over 10 different runs). Results show that our attack outperforms all the model-free attack baselines for the observation manipulation threat model by a large margin for all the statistics. Our proposed attack is also superior on the action manipulation threat model and win over most of the evaluation metrics.

(a) Observation manipulation: mean and standard deviation (in parenthesis)

| Total loss | | **rand-U** | **rand-B** | **flip** | **Ours** |
|---|---|---|---|---|---|
| Walker | stand | 1462 (70) | 1126 (86) | 1458 (24) | **258 (55)** |
| | walk | 1517 (22) | 1231 (31) | 1601 (18) | **466 (42)** |
| Humanoid | stand | 1986 (28) | 1808 (189) | 1997 (5) | **516 (318)** |
| | walk | 1935 (22) | 1921 (31) | 1982 (9) | **1457 (146)** |
| Cartpole | balance | 4000 (0.02) | 3999 (0.04) | 3989 (2) | **2101 (64)** |
| | swingup | 3530 (1) | 3525 (1) | 3516 (1) | **2032 (172)** |

(b) Observation manipulation: averaged ratio and rank-1

| Total loss (avg ratio) | | **Ours/rand-U** | **Ours/rand-B** | **Ours/flip** | **best attack** |
|---|---|---|---|---|---|
| Walker | stand | 0.18 | 0.23 | 0.18 | **Ours: 10/10, others: 0/10** |
| | walk | 0.31 | 0.38 | 0.29 | **Ours: 10/10, others: 0/10** |
| Humanoid | stand | 0.26 | 0.29 | 0.26 | **Ours: 10/10, others: 0/10** |
| | walk | 0.75 | 0.76 | 0.74 | **Ours: 10/10, others: 0/10** |
| Cartpole | balance | 0.53 | 0.53 | 0.53 | **Ours: 10/10, others: 0/10** |
| | swingup | 0.58 | 0.58 | 0.58 | **Ours: 10/10, others: 0/10** |

(c) Action manipulation: mean and standard deviation (in parenthesis)

| Total loss | | **rand-U** | **rand-B** | **flip** | **Ours** |
|---|---|---|---|---|---|
| Cartpole | balance | 4000 (0.03) | 3999 (0.08) | 3046 (1005) | **1917 (102)** |
| | swingup | 3571 (1) | 3487 (7) | 1433 (4) | **1388 (50)** |
| Fish | upright | 935 (27) | 936 (24) | 907 (22) | **824 (84)** |

(d) Action manipulation: averaged ratio and rank-1

| Total loss (avg ratio) | | **Ours/rand-U** | **Ours/rand-B** | **Ours/flip** | **best attack** |
|---|---|---|---|---|---|
| Cartpole | balance | 0.48 | 0.48 | 0.63 | **Ours: 10/10, others: 0/10** |
| | swingup | 0.39 | 0.40 | 0.97 | **Ours: 10/10, others: 0/10** |
| Fish | upright | 0.88 | 0.88 | 0.91 | **Ours: 8/10, flip: 2/10** |

Table 2: Compare three attack baselines (**rand-U**, **rand-B**, **flip**) and our algorithm (**Ours**) in three different domains and tasks. Performance statistics of 10 different runs are reported.

(a) The mean and standard deviation (in parenthesis) over 10 different runs

| Total reward | | **rand-U** | **rand-B** | **flip** | **Ours** |
|---|---|---|---|---|---|
| Walker | stand | 937 (41) | 744 (48) | 993 (8) | **235 (38)** |
| | walk | 941 (23) | 796 (21) | 981 (9) | **225 (50)** |
| Humanoid | stand | 927 (21) | 809 (85) | 959 (5) | **193 (114)** |
| | walk | 934 (22) | 913 (21) | 966 (6) | **608 (66)** |
| Cartpole | balance | 995 (0.17) | 986 (0.16) | 985 (3) | **385 (6)** |
| | swingup | 873 (0.75) | 851 (2) | 852 (0.29) | **353 (61)** |

(b) Average ratio and number of times our algorithm being the best attack over 10 runs.

| Total reward (avg ratio) | | **Ours/rand-U** | **Ours/rand-B** | **Ours/flip** | **best attack** |
|---|---|---|---|---|---|
| Walker | stand | 0.25 | 0.32 | 0.24 | **Ours: 10/10, others: 0/10** |
| | walk | 0.24 | 0.28 | 0.23 | **Ours: 10/10, others: 0/10** |
| Humanoid | stand | 0.21 | 0.24 | 0.20 | **Ours: 10/10, others: 0/10** |
| | walk | 0.65 | 0.67 | 0.63 | **Ours: 10/10, others: 0/10** |
| Cartpole | balance | 0.39 | 0.39 | 0.39 | **Ours: 10/10, others: 0/10** |
| | swingup | 0.41 | 0.42 | 0.42 | **Ours: 10/10, others: 0/10** |

walker and humanoid falls down quickly (head height is close to zero) under our specially-designed attack while remaining unaffected for all the other baselines.

## 4.2 DISCUSSION

**Evaluating on the total reward.** Often times, the reward function is a complicated function and its exact definition is often unavailable. Learning the reward function is also an active research field, which is not in the coverage of this paper. Nevertheless, as long as we have some knowledge of unsafe states (which is often the case in practice), then we can define unsafe states that are related to low reward and thus performing attacks based on unsafe states (i.e. minimizing the total loss of distance to unsafe states) would naturally translate to decreasing the total reward of agent. As demonstrated in Table 2, the results have the same trend of the total loss result in Table 1, where our proposed attack significantly outperforms all the other three baselines. In particular, our method can lower the average total reward up to $4.96\times$ compared to the baselines result, while the baseline results are close to the perfect total reward of 1000.

**Evaluating the effect of planning length.** To investigate model effect over time, we perform ablation studies on the planning/unroll length $T$ of our proposed model-based attack in three examples: (I) cartpole.balance (II) walker.walk and (III) walker.stand.

*(I) Cartpole balance.* Our learned models are very accurate (test MSE error on the order of $10^{-6}$). We observed that the prediction error of our learned model compared to the true model (the MuJoCo simulator) is around 10% for 100 steps. Hence, we can choose $T$ to be very large (e.g. 20-100) and our experiments show that the result of $T = 100$ is slightly better, see Appendix A.4.

*(II) Walker walk.* This task is much more complicated than (I), and our learned model is less accurate (test MSE is $0.447$). For 10 steps, the prediction error of our learned model compared to the true model is already more than 100%, and hence using a small T for planning would be more reasonable. Table 3a shows that $T = 1$ indeed gives the best attack results (decreases the loss by $3.2\times$ and decreases the reward by $3.6\times$ compared to the best baseline (randB)) and the attack becomes less powerful as T increases. Nevertheless, even with $T = 10$, our proposed technique still outperforms the best baseline (randB) by $1.4\times$ both in the total loss and total reward.

Table 3: Ablation study on the planning length $T$. Compare 3 attack baselines (**rand-U**, **rand-B**, **flip**) and our algorithm (**Ours**) and report performance statistics of 10 different runs.

(a) domain: Walker, task: walk (observation perturbation)

| Walker.walk | Total loss | | | | | Total reward | | | | |
|---|---|---|---|---|---|---|---|---|---|---|
| | mean | std | med | min | max | mean | std | med | min | max |
| **Ours,** $T = 1$ | 468 | 79 | 489 | 286 | 567 | 222 | 45 | 227 | 135 | 300 |
| **Ours,** $T = 2$ | 604 | 31 | 611 | 535 | 643 | 353 | 51 | 362 | 253 | 441 |
| **Ours,** $T = 5$ | 761 | 65 | 771 | 617 | 837 | 483 | 60 | 496 | 348 | 540 |
| **Ours,** $T = 10$ | 881 | 68 | 886 | 753 | 975 | 568 | 48 | 579 | 469 | 623 |
| **Ours,** $T = 15$ | 874 | 93 | 891 | 723 | 1002 | 583 | 58 | 604 | 483 | 647 |
| **Ours,** $T = 20$ | 937 | 62 | 950 | 804 | 993 | 634 | 41 | 638 | 559 | 687 |
| **rand-U** | 1517 | 22 | 1522 | 1461 | 1542 | 941 | 23 | 945 | 885 | 965 |
| **rand-B** | 1231 | 31 | 1234 | 1189 | 1272 | 796 | 21 | 796 | 766 | 824 |
| **flip** | 1601 | 18 | 1604 | 1562 | 1619 | 981 | 9 | 984 | 961 | 991 |

(b) domain: Walker, task: stand (observation perturbation)

| Walker.stand | Total loss | | | | | Total reward | | | | |
|---|---|---|---|---|---|---|---|---|---|---|
| | mean | std | med | min | max | mean | std | med | min | max |
| **Ours,** $T = 1$ | 322 | 84 | 319 | 202 | 453 | 257 | 67 | 265 | 163 | 366 |
| **Ours,** $T = 2$ | 279 | 55 | 264 | 223 | 391 | 246 | 40 | 232 | 200 | 322 |
| **Ours,** $T = 5$ | 163 | 53 | 154 | 93 | 246 | 193 | 27 | 188 | 154 | 238 |
| **Ours,** $T = 10$ | 84 | 46 | 67 | 42 | 165 | 153 | 24 | 142 | 132 | 194 |
| **Ours,** $T = 15$ | 101 | 40 | 82 | 57 | 157 | 164 | 23 | 152 | 143 | 201 |
| **Ours,** $T = 20$ | 117 | 41 | 98 | 68 | 193 | 170 | 21 | 161 | 149 | 207 |
| **rand-U** | 1462 | 70 | 1454 | 1341 | 1561 | 938 | 41 | 932 | 866 | 999 |
| **rand-B** | 1126 | 86 | 1130 | 973 | 1244 | 744 | 48 | 744 | 664 | 809 |
| **flip** | 1458 | 24 | 1451 | 1428 | 1501 | 993 | 8 | 997 | 979 | 999 |

Table 4: Less frequency attack. Report statistics of 10 different runs with different initial states in the walker domain with task stand.

| | Total loss | | | | | Total reward | | | | |
|---|---|---|---|---|---|---|---|---|---|---|
| | mean | std | med | min | max | mean | std | med | min | max |
| **Ours** | 934 | 152 | 886 | 769 | 1187 | 648 | 95 | 622 | 559 | 799 |
| **rand-U** | 1511 | 35 | 1502 | 1468 | 1558 | 970 | 20 | 964 | 947 | 999 |
| **rand-B** | 1431 | 77 | 1430 | 1282 | 1541 | 924 | 41 | 923 | 840 | 981 |
| **flip** | 1532 | 15 | 1537 | 1496 | 1546 | 996 | 5 | 999 | 984 | 1000 |

*(III) Walker stand.* The learned model is slightly more accurate than the (II) (test MSE is $0.089$) in this task. Interestingly, Table 3b show that with the more accurate walker.stand model (compared to the walker.walk model), $T = 10$ gives the best avg total loss& reward , which are $13.4\times$ and $4.9\times$ smaller than the best baseline **rand-B**. Note that even with $T = 1$, the worst choice among all our reported $T$, the result is still $3.5\times$ and $2.9\times$ better than the best baselines, demonstrating the effectiveness of our proposed approach.

The main takeaway from these experiments is that when the model is accurate, we can use larger $T$ in our proposed attack; while when the model is less accurate, smaller $T$ is more effective (as in the Walker.walk example). However, even under the most unfavorable hyperparameters, our proposed attack still outperforms all the baselines by a large margin.

**Evaluating on the effectiveness of attack.** We study the setting where attackers are less powerful – they can only attack every 2 time steps instead of every transition. Table 4 shows that our proposed

attack is indeed much stronger than the baselines even when the attackers power is limited to attack every 2 time steps: (1) compared to the best results among three baselines, our attack gives $1.53\times$ smaller avg total loss (2) the mean reward of all the baselines is close to perfect reward, while our attacks can achieve $1.43\times$ smaller average total reward compared to the best baseline.

**Evaluating on the efficiency of attack.** We also study the efficiency of the attack in terms of sample complexity, i.e. how many episodes do we need to perform an effective attack? Here we adopt the convention in control suite (Tassa et al., 2018) where one episode corresponds to $1000$ time steps (samples) and learn the neural network dynamical model $f$ with different number of episodes.

Figure 3 in Appendix A.1 plots the total head height loss of the walker (task stand) for 3 baselines and our method with dynamical model $f$ trained with three different number of samples: $\{5e5, 1e6, 5e6\}$, or equivalently $\{500, 1000, 5000\}$ episodes. We note that the sweep of hyper parameters is the same for all the three models, and the only difference is the number of training samples. The results show that for the baselines **rand-U** and **flip**, the total losses are roughly at the order of 1400-1500, while a stronger baseline **rand-B** still has total losses of 900-1200. However, if we solve Eq. equation 3 with $f$ trained by $5e5$ or $1e6$ samples, the total losses can be decreased to the order of 400-700 and are already winning over the three baselines by a significant margin. Same as our expectation, if we use more samples (e.g. $5e6$, which is 5-10 times more), to learn a more accurate dynamics model, then it is beneficial to our attack method – the total losses can be further decreased by more than $2\times$ and are at the order of 50-250 over 10 different runs. See Appendix A.1 for more details.

Here we also give a comparison between our model-based attack to existing works (Uesato et al., 2018; Gleave et al., 2019) on the sample complexity. In (Uesato et al., 2018), $3e5$ episodes of training data is used to learn the adversarial value function, which is roughly $1000\times$ more data than even our strongest adversary (with $5e3$ episodes). Similarly, (Gleave et al., 2019) use roughly $2e4$ episodes to train an adversary via deep RL, which is roughly $4\times$ more data than ours[2].

## 5 Conclusions and future works

In this paper, we study the problem of adversarial attacks in deep RL with continuous control for two commonly-used threat models. We proposed the first model-based attack algorithm and showed that our formulation can be easily solved by off-the-shelf gradient-based solvers. Extensive experiments on 4 MuJoCo domains show that our proposed algorithm outperforms all model-free based attack baselines by a large margin. We hope our discovery of the vulnerability of deep RL agent can bring more safety awareness to researchers when they design algorithms to train deep RL agents.

There are several interesting future directions can be investigated based on this work, including learning reward functions to facilitate a more effective attack, extending our current approach to develop effective black-box attacks, and incorporating our proposed attack algorithm to adversarial training of the deep RL agents. In particular, we think there are three important challenges that need to be addressed to study adversarial training of RL agents along with our proposed attacks: (1) The adversary and model need to be jointly updated. How do we balance these two updates, and make sure the adversary is well-trained at each point in training? (2) How to avoid cycles in the training process due to the agent overfitting to the current adversary? (3) How to ensure the adversary doesn't overly prevent exploration/balance unperturbed vs. robust performance?

## Acknowledgement

The authors thank Chongli Qin, Po-Sen Huang, Taylan Cemgil, Daniel J. Mankowitz, Nir Levine, Alistair Muldal, Gabriel Barth-Maron, Matthew W. Hoffman, Yuval Tassa, Tom Erez and Jost Tobias Springenberg for useful discussions and suggestions.

---

[2]It is only a qualitative comparison in the sample complexity regimes since the applications are not the same. This is in-line with the theoretical perspective that model-based approaches are expected to be more sample-efficient than the model-free counterparts

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

# A APPENDIX

## A.1 MORE ILLUSTRATION ON FIGURE 3

The meaning of Fig 3 is to show how the accuracy of the learned models affects our proposed technique:

1. we first learned 3 models with 3 different number of samples: 5e5, 1e6, 5e6 and we found that with more training samples (e.g. 5e6, equivalently 5000 episodes), we are able to learn a more accurate model than the one with 5e5 training samples;

2. we plot the attack results of total loss for our technique with 3 learned models (denoted as PGD, num_train) as well as the baselines (randU, randB, Flip) on 10 different runs (initializations).

We show with the more accurate learned model (5e6 training samples), we are able to achieve a stronger attack (the total losses are at the order of 50-200 over 10 different runs) than the less accurate learned model (e.g. 5e5 training samples). However, even with a less accurate learned model, the total losses are on the order of 400-700, which already outperforms the best baselines by a margin of 1.3-2 times. This result in Fig 3 also suggest that a very accurate model isn't necessarily needed in our proposed method to achieve effective attack. Of course, if the learned model is more accurate, then we are able to degrade agent's performance even more.

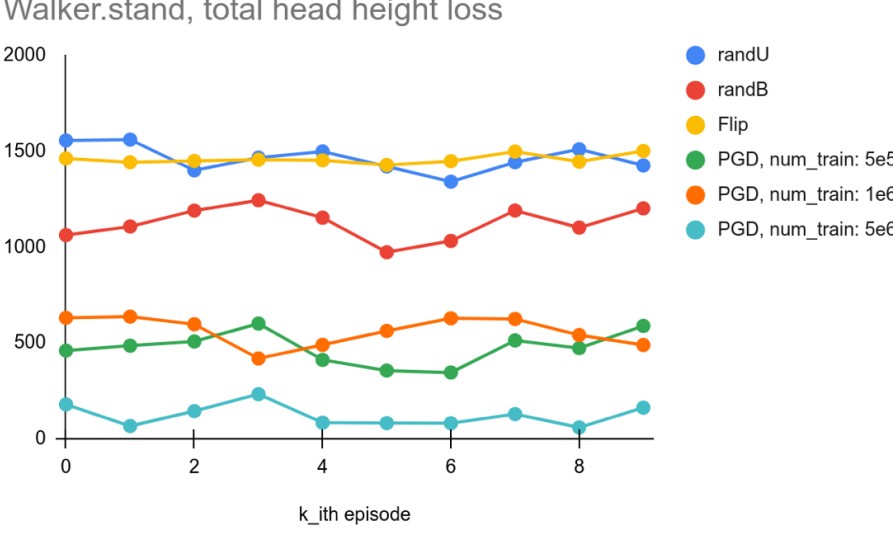

Figure 3: Compare sample size on the Walker.stand in 10 different initialization in the environment. The x-axis is the $k$th initialization and the y-axis is the total loss of corresponding initialization.

## A.2 MORE DETAILS ON BASELINE IMPLEMENTATIONS

For the baselines (rand-U and rand-B), the adversary generates 1000 trajectories with random noise directly and we report the best loss/reward at the end of each episode. The detailed steps are listed below:

Step 1: The perturbations are generated from a uniform distribution or a bernoulli distribution within the range [-eps, eps] for each trajectory, and we record the total reward and total loss for each trajectory from the true environment (the MuJoCo simulator)

Step 2: Take the best (lowest) total reward/loss among 1000 trajectories and report in Table 1 and 2.

We note that here we assume the baseline adversary has an "unfair advantage" since they have access to the true reward (and then take the best attack result among 1000 trials), whereas our techniques do not have access to this information. Without this advantage, the baseline adversaries (rand-B, rand-U) may be weaker if they use their learned model to find the best attack sequence. In any case, Table 1 and 2 demonstrate that our proposed attack can successfully uncover vulnerabilities of deep RL agents while the baselines cannot.

For the baseline **flip**, we add the perturbation (with the opposite sign and magnitude $\epsilon$) on the original state/action and project the perturbed state/action are within its limits.

## A.3    SCORE OF LEARNED POLICY WITHOUT ATTACKS

We use default total timesteps = 1000, and the maximum total reward is 1000. We report the total reward of the d4pg agents used in this paper below. The agents are well-trained and have total reward close to 1000, which outperforms agents trained by other learning algorithms on the same tasks (e.g. DDPG, A3C in Sec 6 (Tassa et al., 2018); PPO in Sec 5 (Abdolmaleki et al., 2018)), and thus the agents in this paper can be regarded as state-of-the-art RL agents for these continuous control domain tasks. The attack results in Table 1 and 2 in our manuscript are hence suggested to be representative.

| Domain | Task | Total reward |
|---------|---------|------|
| Walker | stand | 994 |
| Walker | walk | 987 |
| Humanoid | stand | 972 |
| Humanoid | walk | 967 |
| Cartpole | balance | 1000 |
| Cartpole | swingup | 883 |
| Fish | upright | 962 |

## A.4    ADDITIONAL EXPERIMENTS ON ABLATION STUDY

Table 5: Cartpole balance (action perturbation)

| Total loss | mean | std | med | min | max |
|------------|------|------|------|------|------|
| **Ours,** $T = 20$ | 2173 | 51 | 2189 | 2087 | 2239 |
| **Ours,** $T = 100$ | 1951 | 113 | 1924 | 1851 | 2192 |
| **rand-U** | 4000 | 0 | 4000 | 4000 | 4000 |
| **rand-B** | 3999 | 0 | 3999 | 3999 | 3999 |
| **flip** | 3046 | 1005 | 3074 | 2060 | 3999 |

