# OpenReview forum: "Toward Evaluating Robustness of Deep Reinforcement Learning with Continuous Control"
_ICLR.cc/2020/Conference — Accept (Poster)_

### Official Review · AnonReviewer2 · 2019-10-20
**Official Blind Review #2**

**Rating:** 6

**Review:**

Summary: This paper proposed a new adversarial attack method based on model-based RL. Unlike existing adversarial attack methods on deep RL, the authors first approximate the dynamics models and then generate the adversarial samples by minimizing the total distance of each state to the pre-defined target state (i.e. planning). Using Cartpole, Fish, Walker, and Humanoid, the authors showed that the proposed method can pool the agents more effectively.

Detailed comments:

The proposed idea (i.e. designing an adversarial attack based on model-based RL) is interesting but it would be better if the authors can provide evaluations such as adversarial training and ablation studies for the proposed method (see the suggestion & question). So, I'd like to recommend "weak accept"

Suggestion & question:

Could the authors apply adversarial training based on the proposed methods? I wonder whether RL agents can be robust after adversarial training.

Instead of utilizing a pre-defined target state $s_{target}$, we can also approximate a reward function and generate adversarial samples by minimizing the total rewards. It would be interesting if the authors can consider this case.

Could the authors report an ablation study on the effects of T?

**Experience Assessment:**

I have published one or two papers in this area.

**Review Assessment: Checking Correctness Of Derivations And Theory:**

I carefully checked the derivations and theory.

**Review Assessment: Checking Correctness Of Experiments:**

I carefully checked the experiments.

**Review Assessment: Thoroughness In Paper Reading:**

I read the paper thoroughly.

---

> ### Author Response · Authors · 2019-11-15
> **Reply to Reviewer 2, part 1**
>
> Thanks for your positive feedback!
>
> #1  adversarial training on RL agents
> Yes, we believe this is definitely a very interesting research direction. From our point of view, we think there are three important challenges that need to be addressed to study adversarial training of RL agents along with our proposed attacks:
>
> 1. The adversary and model need to be jointly updated. How do we balance these two updates, and make sure the adversary is well-trained at each point in training?
> 2. How do we avoid cycles in the training process due to the agent overfitting to the current adversary?
> 3. How do we ensure the adversary doesn't overly prevent exploration / balance unperturbed vs. robust performance?
>
> While we think adversarial training may help to train a more robust agent in general (similar to its image classification counterpart [Madry et. al 2018]), we believe it would be better to study this question systematically and rigorously in a separate paper due to the above challenges and rebuttal time constraints. We hope to focus this paper on the robustness evaluation of deep RL agent because our goal is to uncover model vulnerability in the field of deep RL -- to bring researchers awareness of the potential safety issues of state-of-the-art RL agents -- and we hope our results can better motivate adversarial training as an important next step to help to train more robust deep RL agents. We have updated our manuscript to discuss the importance of this direction and identify some of the challenges and nuances in the future work section in the appendix.
>
> Reference:
> - Madry et. al, Towards Deep Learning Models Resistant to Adversarial Attacks, ICLR 2018
>
> #2  ablation study on T
> Following your suggestion, we perform ablation studies on T (the planning length) of our proposed model-based attack in three examples (walker.stand, walker.walk, cartpole.balance) and conduct experiments for 10 different runs with different initial states. The following statistics of 10 runs (mean, standard deviation, median, min and max) are reported.
>
> The main takeaway from these experiments is that when the model is accurate, we can use larger T in our proposed attack algorithm; while when the model is less accurate, shorter unrolls (smaller T) are more effective (as in the Walker.walk example). However, even under the most unfavorable hyperparameters, our proposed attack still outperforms all the baselines by a large margin. For example, using the least accurate model with T=20 in Walker.stand example, decreases the model’s avg reward to 170 compared to 744 for the best baseline. In general, we suggest choosing a moderate T that can leverage the learned model while remaining robust to errors in the learned model.
>
> Our results:
> 1. domain: Walker, task: stand (observation perturbation), test MSE:
> (a) Total loss (smaller means stronger attack)
>                 	mean    std    med     min     max
> Ours, T=1     	  322      84     319       202      453
> Ours, T=2   	  279      55     264       223      391
> Ours, T=5   	  163      53     154         93      246
> Ours, T=10  	    84      46       67         42      165
> Ours, T=15   	  101      40       82         57      157
> Ours, T=20   	  117      41       98         68      193
> randU    	        1462      70    1454    1341    1561
> randB    	        1126      86    1130      973    1244
> flip         	        1458      24    1451    1428    1501
>
> (b) Total reward (smaller means stronger attack)
>                 	mean    std    med     min     max
> Ours, T=1     	  257       67     265      163      366
> Ours, T=2   	  246       40     232      200      322
> Ours, T=5   	  193       27     188      154      238
> Ours, T=10  	  153       24     142      132      194
> Ours, T=15   	  164       23     152      143      201
> Ours, T=20   	  170       21     161      149      207
> randU    	          938       41     932      866      999
> randB    	          744       48     744      664      809
> flip         	          993         8     997      979      999
>
> (to be continued in part 2)

---

> > ### Author Response · Authors · 2019-11-15
> > **Reply to Reviewer 2, part 2**
> >
> >
> > (continued response to #2 ablation study on T)
> >
> > 2. domain: Walker, task: walk (observation perturbation)
> >
> > (a) Total loss (smaller means stronger attack)
> >                 	mean    std    med    min      max
> > Ours, T=1     	  468      79     489       286      567
> > Ours, T=2   	  604      31     611       535      643
> > Ours, T=5   	  761      65     771       617      837
> > Ours, T=10  	  881      68     886       753      975
> > Ours, T=15   	  874      93     891       723     1002
> > Ours, T=20   	  937      62     950       804      993
> > randU    	        1517      22    1522    1461    1542
> > randB    	        1231      31    1234    1189    1272
> > flip         	        1601      18    1604    1562    1619
> >
> > (b) Total reward (smaller means stronger attack)
> >                 	mean   std    med    min     max
> > Ours, T=1     	  222      45     227     135      300
> > Ours, T=2   	  353      51     362     253      441
> > Ours, T=5   	  483      60     496     348      540
> > Ours, T=10  	  568      48     579     469      623
> > Ours, T=15   	  583      58     604     483      647
> > Ours, T=20   	  634      41     638     559      687
> > randU    	          941      23     945     885      965
> > randB    	          796      21     796     766      824
> > flip         	          981        9     984     961      991
> >
> > 3. domain: Cartpole, task: balance (action perturbation)
> >
> > (a) Total loss (smaller means stronger attack)
> >                 	      mean     std    med     min     max
> > Ours, T=20     	2173       51   2189    2087    2239
> > Ours, T=100   	1951     113   1924    1851    2192
> > randU    	               4000        0     4000    4000    4000
> > randB    	               3999        0     3999    3999    3999
> > flip         	               3046    1005   3074    2060    3999
> >
> >
> > #3 learning reward functions
> > Thanks for your insight! Yes, it is indeed an interesting point, which we also discussed in Sec 4.2 of original manuscript. While we believe learning a surrogate of reward function can help to lower the true reward during PGD attack, it is beyond the scope of this paper. We have added a paragraph of future works in appendix and include this approach as an interesting future work.

---

### Official Review · AnonReviewer3 · 2019-10-22
**Official Blind Review #3**

**Rating:** 3

**Review:**

  *Synopsis*:
  This paper looks at a new framework for adversarial attacks on deep reinforcement learning agents under continuous action spaces. They propose a model based approach which adds noise to either the observation or actions of the agent to push the agent to predefined target states. They then report results against several model-free/unlearned baselines on MuJoCo tasks using a policy learned through D4PG.

  Main contributions:
  - Adversarial attacks for Deep RL in continuous action spaces.

  *Review*
  The paper is well written, and has some interesting discussion/insight into attacking deep RL agents in continuous actions spaces. I think the authors are headed in the right direction, but compared to prior work in adversarial attacks for deep RL agents (i.e. the Huang and Lin) I have a few concerns that I feel the authors need to better explain/motivate in their paper. I am recommending this paper be rejected based on the following concerns. I am willing to raise my score if some of these are addressed by the authors in subsequent revisions

  1. This algorithm requires the pre-trained policy to plan attacks (which may be a high bar for such an adversarial attack). It would be a nice addition to include similar results with "black-box" adversarial attacks, as mentioned in the Huang.

  2. Another issue, addressed in the Lin paper, is this attack seems to require perturbation on every time step in a proposed trajectory. As mentioned by Lin, this is probably unrealistic and would cause the attacker to be detected. It would be another nice contribution to include variants that don't require perturbations on each transition.

  3. Another unfortunate requirement is a learned model (or a way to simulate trajectories). From the Model Based RL literature, we know learning such a model is quite difficult and often unrealistic given our current approaches. While this is problematic, I think the paper could systematically test this looking at what happens as the model becomes less accurate over time. This could provide some nice results showing an accurate model isn't necessarily needed and anneal concerns over having to learn such a model.

  4. It is unclear if the baselines measured against are meaningful in this setting, and I'm also a bit unclear how they are generated/implemented. Specifically, the random trajectories require you to return the generated trajectory with the smallest loss/reward. It is unclear how the adversary knows this information. Is it known through a model or some other simulation? Also the flip baseline could use a bit more explanation. I think these details can be safely placed in the appendix, but should appear somewhere in the final version.

  5. I'm not sure the comparison to sample efficiency to the Gleave or Uesato papers are meaningful. For Gleave, the threat model explored is much different where they do not have access to the agent's observation or action streams and instead learn policies to affect the other agent in game scenarios. This is very different. Also, the Uesato is not adversarially attacking the agent, but attempting to find failure cases for the agent, which I again feel is very different from what you are trying to accomplish. I would remove this discussion and the claim at the end of the conclusion.


  Other suggestions:

  S1. It would be helpful to include the score of the learned policy without any attacks, to see how well the baselines are performing (this will help readers understand if these are reasonable/meaningful baselines).

  S2. I'm unclear what figure three is adding to the paper, and am actually uncertain what the y-axis means. I don't think this is a wise use of the 9th page, and this plot could probably be relegated to the appendix.

  S3. As in prior work, it would be useful to see how well this line of attack works for multiple learning algorithms. Some potential candidates could be: PPO, TRPO, SAC, etc...


**Experience Assessment:**

I have read many papers in this area.

**Review Assessment: Checking Correctness Of Derivations And Theory:**

I assessed the sensibility of the derivations and theory.

**Review Assessment: Checking Correctness Of Experiments:**

I carefully checked the experiments.

**Review Assessment: Thoroughness In Paper Reading:**

I read the paper at least twice and used my best judgement in assessing the paper.

---

> ### Author Response · Authors · 2019-11-15
> **Reply to Reviewer 3, part 1**
>
> Thanks for your detailed response and constructive suggestions to help us improve the original manuscript! We have incorporated all your comments accordingly into the revised manuscript and we hope our additional experimental results and clarification/explanations can convince you about the contributions of this paper.
>
> #1 black-box attacks
> Thanks for your insight! Yes, our proposed technique is a white-box attack, which requires the information of pre-trained agents. The setting of black-box attacks in Huang et. al is interesting but is not the main focus of this paper. We have added a remark and included the black-box attack as an interesting direction in our future work section in the appendix.
>
> #2 attacks with less frequency
> Following your suggestion, we have performed additional experiments on the setting where attackers are less powerful -- they can only attack every 2 timestep instead of every transition. We conduct experiments for 10 different runs with different initial states in the walker domain with task stand and report the following statistics of 10 runs (mean, standard deviation, median, min and max).
>
> The results show that our proposed attack is indeed much stronger than the baselines even when the attacker’s power is limited (e.g. can only attack every 2 timesteps):
> 1. Compared to the best results among three baselines, our attack gives 1.53X smaller avg total loss (ours: 934 v.s. best baseline: 1431)
> 2. The mean reward of all the baselines is close to perfect reward, while our attacks can achieve 1.43X smaller avg total reward compared to the best avg reward from baseline (ours: 648 v.s. best baseline: 924).
>
> (a) Total loss (smaller means stronger attack)
>               mean    std    med    min      max
> Ours       934      152     886     769     1187
> randU    1511      35    1502   1468    1558
> randB    1431      77    1430   1282    1541
> flip         1532      15    1537   1496    1546
>
> (b) Total reward (smaller means stronger attack)
>               mean    std    med    min     max
> Ours       648       95     622      559      799
> randU     970       20     964     947      999
> randB     924       41     923     840      981
> flip          996         5     999     984     1000
>
>
> #3 systematically test what happens as the model becomes less accurate over time
>
> Thanks for your insights. Yes, we have some discussion on the effect of model accuracy in the Sec 4.2 (Evaluating on the efficiency of attack) and Fig 3 in our original manuscript. We show that even with a less accurate learned model (with only 5e5 training samples, equivalently 500 episodes), our proposed attack can already successfully degrade the agent performance by a factor of 1.3-2X compared to the best baseline results. This suggests that an accurate model isn't necessarily needed in our proposed method to achieve effective attacks -- note that with a more accurate learned model (with 5e6 training samples), our proposed attack can further degrade the agent performance by 2X. Please also see our reply #7 Figure 3 clarification.
>
> In addition, to investigate model effect over time, we have performed additional experiments on the planning/unroll length T for 3 example domains and tasks, each with 10 different initializations. The results are summarized below.
>
> 1. domain: Cartpole, task: balance (action perturbation)
> For the Cartpole balance task, our learned models are very accurate (test MSE error on the order of 1e-6). We observed that the prediction error of our learned model compared to the true model (the MuJoCo simulator) is around 10% for 100 steps. Hence, we can choose the planning steps T to be very large (e.g. 20-100) and our experiments show that the result of T = 100 is slightly better:
>
> (a) Total loss (smaller means stronger attack)
>                 	      mean     std    med     min     max
> Ours, T=20     	2173       51   2189    2087    2239
> Ours, T=100   	1951     113   1924    1851    2192
> randU    	               4000        0     4000    4000    4000
> randB    	               3999        0     3999    3999    3999
> flip         	               3046    1005   3074    2060    3999
>
>
> (to be continued in part 2)

---

> > ### Author Response · Authors · 2019-11-15
> > **Reply to Reviewer 3, part 2**
> >
> >
> > (continued response of #3 systematically test what happens as the model becomes less accurate over time)
> >
> > 2. domain: Walker, task: walk (observation perturbation)
> > For Walker walk task, on the other hand, it is much more complicated, and our learned models are less accurate (test error on the order of 1e-1). For 10 steps, the prediction error of our learned model compared to the true model is already more than 100%, and hence using a small T for planning would be more reasonable. We report the results for T = {1,2,5,10,15,20} over 10 runs below and compare the total loss and total reward with the baseline results.
> >
> > Our results show that using T = 1 indeed gives the best attack results (decreases the loss by 3.2X and decreases the reward by 3.6X compared to the best baseline (randB)) and the attack becomes less powerful as T increases. Nevertheless, even with T = 10, our proposed technique still outperforms the best baseline (randB) by 1.4X both in the total loss and total reward.
> >
> > (a) Total loss (smaller means stronger attack)
> >                 	mean    std    med    min      max
> > Ours, T=1     	  468      79     489       286      567
> > Ours, T=2   	  604      31     611       535      643
> > Ours, T=5   	  761      65     771       617      837
> > Ours, T=10  	  881      68     886       753      975
> > Ours, T=15   	  874      93     891       723     1002
> > Ours, T=20   	  937      62     950       804      993
> > randU    	        1517      22    1522    1461    1542
> > randB    	        1231      31    1234    1189    1272
> > flip         	        1601      18    1604    1562    1619
> >
> > (b) Total reward (smaller means stronger attack)
> >                 	mean   std    med    min     max
> > Ours, T=1     	  222      45     227     135      300
> > Ours, T=2   	  353      51     362     253      441
> > Ours, T=5   	  483      60     496     348      540
> > Ours, T=10  	  568      48     579     469      623
> > Ours, T=15   	  583      58     604     483      647
> > Ours, T=20   	  634      41     638     559      687
> > randU    	          941      23     945     885      965
> > randB    	          796      21     796     766      824
> > flip         	          981        9     984     961      991
> >
> > 3. domain: Walker, task: stand (observation perturbation)
> > For Walker stand, the learned model is slightly more accurate than the walker.walk model (we use the test MSE error for evaluation, the test MSE for walker.walk is 0.447 while the test MSE for walker.stand is 0.089), and we also study the effect of T on our proposed method over 10 runs.
> >
> > Interestingly, our results show that with the more accurate walker.stand model (compared to the walker.walk model), T = 10 gives the best avg total loss (84) and best avg total reward (153), which are 13.4X and 4.9X smaller than the best baseline, randB (avg total loss: 1126, avg total reward: 744). Again, note that even with T = 1, the worst choice among all T = {1,2,5,10,15,20}, the result is still 3.5X and 2.9X better than the best baselines, demonstrating the effectiveness of our proposed approach.
> >
> > (a) Total loss (smaller means stronger attack)
> >                 	mean    std    med     min     max
> > Ours, T=1     	  322      84     319       202      453
> > Ours, T=2   	  279      55     264       223      391
> > Ours, T=5   	  163      53     154         93      246
> > Ours, T=10  	    84      46       67         42      165
> > Ours, T=15   	  101      40       82         57      157
> > Ours, T=20   	  117      41       98         68      193
> > randU    	        1462      70    1454    1341    1561
> > randB    	        1126      86    1130      973    1244
> > flip         	        1458      24    1451    1428    1501
> >
> > (b) Total reward (smaller means stronger attack)
> >                 	mean    std    med     min     max
> > Ours, T=1     	  257       67     265      163      366
> > Ours, T=2   	  246       40     232      200      322
> > Ours, T=5   	  193       27     188      154      238
> > Ours, T=10  	  153       24     142      132      194
> > Ours, T=15   	  164       23     152      143      201
> > Ours, T=20   	  170       21     161      149      207
> > randU    	          938       41     932      866      999
> > randB    	          744       48     744      664      809
> > flip         	          993         8     997      979      999
> >
> >
> > #4 Baseline implementation
> > 1. For the baselines (rand-U and rand-B), the adversary generates 1000 trajectories with random noise directly and we report the best loss/reward at the end of each episode. The detailed steps are listed below:
> >
> > Step 1: the perturbations are generated from a uniform distribution or a bernoulli distribution within the range [-eps, eps] for each trajectory, and we record the total reward and total loss for each trajectory from the true environment (the MuJoCo simulator).
> >
> > Step 2: take the best (lowest) total reward/loss among 1000 trajectories and report in Table 1 and 2 in our original manuscript.
> >
> > (to be continued in part 3)

---

> > > ### Author Response · Authors · 2019-11-15
> > > **Reply to Reviewer 3, part 3**
> > >
> > >
> > > (continued response to #4 Baseline implementation)
> > >
> > > 2. Yes, the reviewer's interpretation is correct -- we assume the baseline adversary has an "unfair advantage" since they have access to the true reward (and then take the best attack result among 1000 trials), whereas our techniques do not have access to this information. Without this advantage, the baseline adversaries (rand-B, rand-U) may be weaker if they use their learned model to find the best attack sequence. In any case, Table 1 and 2, as well as the above additional experiments (#2 less frequency attack, #3 learned model) all demonstrate that our proposed attack can successfully uncover vulnerabilities of deep RL agents while the baselines cannot.
> > >
> > > Following your suggestion, we have added the above details to the appendix.
> > >
> > >
> > > #5 Sample complexity comparison
> > >
> > > Thanks for your comment! Yes, we agree the applications/setting are different, and our point was to highlight a qualitative difference in the sample complexity regimes (ours ~1000 vs Gleave et. al ~100K or Uesato et. al >1M), even though these clearly aren't head-to-head comparisons. This is in-line with the theoretical perspective that we might expect model-based approaches to be more sample-efficient than their model-free counterparts. Following your suggestion, we have added a footnote to clarify that this is a qualitative comparison, and we have also removed the claim at the end of the conclusion.
> > >
> > >
> > > #6 Score of learned policy without attacks and other learning algorithms
> > >
> > > We use default total timesteps = 1000, and the maximum total reward is 1000. We report the total reward of the d4pg agents used in this paper below. The agents are well-trained and have total reward close to 1000, which outperforms agents trained by other learning algorithms on the same tasks (e.g. ddpg, A3C in sec 6, Tassa et. al 2018; ppo in sec 5, Abdolmaleki et. al 2018), and thus the agents in this paper can be regarded as state-of-the-art RL agents for these continuous control domain tasks. The attack results in Table 1 and 2 in our manuscript are hence suggested to be representative.
> > >
> > > Domain         	Task           Total reward
> > > walker	        stand              994
> > > walker           	walk               	987
> > > humanoid     stand             	972
> > > humanoid     walk               	967
> > > cartpole        	balance         1000
> > > cartpole        	swingup         883
> > > fish               	upright           962
> > >
> > > Reference:
> > > - Tassa et. al 2018, Deepmind control suite.
> > > - Abdolmaleki et. al 2018, Maximum A Posteriori Policy Optimisation.
> > >
> > >
> > > #7 Figure 3 clarification
> > >
> > > Thanks for your suggestion, we have moved Fig 3 to the appendix. The y-axis is the total head height loss (originally described in the title of figure as well as in the caption "y-axis is the total loss of the corresponding initialization"), and the x-axis is the k_th run. As discussed in Sec 4.2, the meaning of Fig 3 is to show how the accuracy of the learned models affects our proposed technique:
> > >
> > > (1) we first learned 3 models with 3 different number of samples: {5e5, 1e6, 5e6} and we found that with more training samples (e.g. 5e6, equivalently 5000 episodes), we are able to learn a more accurate model than the one with 5e5 training samples.
> > >
> > > (2) we plot the attack results of total loss for our technique with 3 learned models (denoted as PGD, num_train) as well as the baselines (randU, randB, Flip) on 10 different runs (initializations).
> > >
> > > We show with the more accurate learned model (5e6 training samples), we are able to achieve a stronger attack (the total losses are at the order of 50-200 over 10 different runs) than the less accurate learned model (e.g. 5e5 training samples). However, even with a less accurate learned model, the total losses are on the order of 400-700, which already outperforms the best baselines by a margin of 1.3-2 times. This result in Fig 3 also answers reviewer's comments #3 that indeed, to achieve effective attack, a very accurate model isn't necessarily needed in our proposed method. Of course, if the learned model is more accurate, then we are able to degrade agent's performance even more.
> > >
> > >
> > > We appreciate your constructive feedback and hope our additional experimental results and clarification can convince you about the contributions of this paper.

---

> > > > ### Author Response · Authors · 2019-11-15
> > > > **Additional experiment to Reviewer 3 #1**
> > > >
> > > >
> > > > #1 black-box attacks
> > > >
> > > > Following your suggestion, we managed to perform an additional experiment on the black-box transfer attack in walker domain with task stand with planning length T = 5. The experiment is conducted in 9 different runs and we report the avg total reward and avg total loss below.
> > > >
> > > > Without any attacks, Agent 1 has total reward of 995 and the Agent 2 has total reward of 994. Agent 1 and 2 are both d4pg agents with the same architecture but different network parameters.
> > > >
> > > > * If we perform white box attack on the Agent 1 in 9 different runs, then the mean of total loss is 172 with std 59, and the mean of total reward is 197 with std 28.
> > > >
> > > > * If we perform black box attack on the Agent 2, using the perturbations generated from Agent 1 (since under the black-box assumption, we don't have the parameters of targeted Agent 2), the mean of total loss is 1550 with std 47 and the mean of total reward is 985 with std 20.
> > > >
> > > > It can be seen that the transfer attack is not as effective as our white box attack on Agent 1. A possible reason could be that the two Agents may be very different despite both of them perform well in the normal setting without attacks. Hence, we believe the black-box setting would certainly be an interesting future direction that requires a more comprehensive study and investigation.

---

### Official Review · AnonReviewer1 · 2019-10-25
**Official Blind Review #1**

**Rating:** 6

**Review:**

This paper presents an adversarial attack for perturbing
the actions or observations of an agent acting near-optimally
in an MDP so that the policy performs poorly.
I think understanding the sensitivity of a policy to
slight perturbations in the actions it takes or the
observations that it receives is important for having
robust learned policies and controllers.
This paper presents an empirical step in the direction
of showing that such attacks are possible, but in the context
of the other adversarial attacks that are possible, this is
not surprising alone and would be much stronger with
other contributions.
I think an exciting direction of new work could be to
continue formalizing these vulnerabilities and
looking at ways of adding robustness across many
other domains.

**Experience Assessment:**

I have published one or two papers in this area.

**Review Assessment: Checking Correctness Of Derivations And Theory:**

I assessed the sensibility of the derivations and theory.

**Review Assessment: Checking Correctness Of Experiments:**

I assessed the sensibility of the experiments.

**Review Assessment: Thoroughness In Paper Reading:**

I made a quick assessment of this paper.

---

> ### Author Response · Authors · 2019-11-15
> **Reply to Reviewer 1**
>
> Thanks for your comments!
>
> #1 Other attacks are possible
> Indeed, adversarial attacks have been shown to be possible for neural network models in various supervised learning tasks, and hence it may not be surprising that adversarial attacks exist for the deep RL agents.
>
> However, the vulnerability of RL agents can not be easily discovered by existing baselines which are model-free and build upon random searches and heuristics -- this is also verified by our extensive experiments on various domains (e.g. walker, humanoid, cartpole, and fish), where the agents still achieve close to their original best rewards even with baseline attacks at every time step (see Table 1 and 2).
>
> Hence, it is important and necessary to have a systematic methodology to design *non-trivial* adversarial attacks, which can *efficiently* and *effectively* discover the vulnerabilities of deep RL agents -- this is indeed the motivation of this work. This paper takes a first step toward this direction by proposing the first sample-efficient model-based adversarial attack, which can successfully degrade agent performance by up to 4 times in terms of total reward and up to 4.6 times in terms of distance to unsafe states (smaller means stronger attacks) compared to the model-free baselines.
>
> Therefore, we believe (1) our proposal of non-trivial model-based adversarial attacks and (2) our systematic study on the efficiency and effectiveness of our method compared with model-free attack baselines are two important contributions to the field of deep reinforcement learning. We hope our discovery of the vulnerability of deep RL agent can also bring more safety awareness to researchers in this field when they design algorithms to train deep RL agents.
>
> #2 Our additional contributions
> In the rebuttal, we have conducted additional experiments to further demonstrate the effectiveness of our proposed method and show that our techniques outperform all the baselines by a significant margin:
> (1) In a weaker adversary setting where the adversary cannot attack at every time step, we show that our attack can still successfully degrade agent's performance (total reward and loss) by 1.4-1.5X while the baseline attacks cannot. Note that the agents are almost not affected by the baseline attacks because the agents still achieve almost perfect total reward. Details please see our reply #2 to Reviewer#3.
>
> (2) We conduct systematic study in three domains and tasks on the effect of planning/unroll length in our proposed technique. Our results suggest that our technique are still effective and outperforms all the baselines by a large margin even when the learned model dynamics is not very accurate. In particular, we have discussed the trade-off between unroll length T and the model accuracy in three examples. Details please see our reply #2 to Reviewer #2 and reply #3, #7 to Reviewer #3.
>
> We hope that the above explanation and clarification have convinced you of the importance and contributions of this paper.

---

> > ### Comment · AnonReviewer1 · 2019-11-15
> > **Reviewer response**
> >
> > Thanks for the response and clarifications, especially for the detailed response to R3. I have increased my score to a weak accept as I agree that this is a relevant and important topic to study and that the experiments shown in this paper are novel in some way -- however I am still feeling borderline about the paper as it seems like it is set up in a way that causes the baselines to trivially fail giving the existing body of adversarial attack literature.

---

### Decision · Program_Chairs · 2019-12-19

**Decision:**

Accept (Poster)

**Comment:**

This paper considers adversarial attacks in continuous action model-based deep reinforcement learning. An optimisation-based approach is presented, and evaluated on Mujoco tasks.

There were two main concerns from the reviewers. The first was that the approach requires strong assumptions, but in the rebuttal some relaxations were demonstrated (e.g., not attacking every step). Additionally, there were issues raised with the choice of baselines, but in the discussion the reviewers did not agree on any other reasonable baselines to use.

This is a novel and interesting contribution nonetheless, which could open the field to much additional discussion, and so should be accepted.